# PlA2 Polymorphism of Platelet Glycoprotein IIb/IIIa and C677T Polymorphism of Methylenetetrahydrofolate Reductase (*MTHFR*), but Not Factor V Leiden and Prothrombin G20210A Polymorphisms, Are Associated with More Severe Forms of Legg–Calvé–Perthes Disease

**DOI:** 10.3390/children8070614

**Published:** 2021-07-20

**Authors:** María Dolores García-Alfaro, María Isabel Pérez-Nuñez, María Teresa Amigo, Carmelo Arbona, María Ángeles Ballesteros, Domingo González-Lamuño

**Affiliations:** 1Department of Orthopaedic Surgery, Hospital Universitario Marqués de Valdecilla, Avda Valdecilla s/n, 39008 Santander, Spain; dolores.garcia@scsalud.es (M.D.G.-A.); perezmi@unican.es (M.I.P.-N.); carmelo.arbona@scsalud.es (C.A.); 2Laboratorio de Pediatría, Departamento de Ciencias Médicas y Quirúrgicas, Universidad de Cantabria, Cardenal Herrera Oria s/n, 39011 Santander, Spain; teresa.amigo@unican.es; 3Department of Critical Care Medicine, Hospital Marqués de Valdecilla-IDIVAL, Avda Valdecilla s/n, 39008 Santander, Spain; mdelosangeles.ballesteros@scsalud.es; 4Division of Pediatrics, Hospital Universitario Marqués de Valdecilla-IDIVAL, Avda Valdecilla s/n, 39008 Santander, Spain

**Keywords:** Legg–Calvé–Perthes disease, gene polymorphism, heritable thrombophilia

## Abstract

The possible association of common polymorphic variants related to thrombophilia (the rs6025(A) allele encoding the Leiden mutation, rs1799963(A), i.e., the G20210A mutation of the prothrombin *F2* gene, the rs1801133(T) variant of the methylenetetrahydrofolate reductase (*MTHFR)* gene that encodes an enzyme involved in folate metabolism, and rs5918(C), i.e., the ‘A2’ allele of the platelet-specific alloantigen system that increases platelet aggregation induced by agonists), with the risk of Legg–Calvé–Perthes disease (LCPD) and the degree of hip involvement (Catterall stages I to IV) was analyzed in a cohort study, including 41 children of ages 2 to 10.9 (mean 5.4, SD 2.2), on the basis of clinical and radiological criteria of LCPD. In 10 of the cases, hip involvement was bilateral; thus, a total of 51 hips were followed-up for a mean of 75.5 months. The distribution of genotypes among patients and 118 controls showed no significant differences, with a slightly increased risk for LCPD in rs6025(A) carriers (OR: 2.9, CI: 0.2–47.8). Regarding the severity of LCPD based on Catterall classification, the rs1801133(T) variant of the *MTHFR* gene and the rs5918(C) variant of the platelet glycoprotein IIb/IIIa were associated with more severe forms of Perthes disease (Catterall III–IV) (*p* < 0.05). The four children homozygous for mutated *MTHFR* had a severe form of the disease (Stage IV of Catterall) and a higher risk of non-favorable outcome (Stulberg IV–V).

## 1. Introduction

Legg–Calvé–Perthes disease (LCPD), characterized by avascular necrosis of the proximal femoral epiphysis [1,2,3,4], remains a controversial entity in child orthopedics. During rapid growth, the interruption of adequate blood supply to the secondary ossification centers in the epiphyses makes these areas prone to avascular necrosis, resulting in necrosis, removal of the necrotic tissue, and its replacement with new bone [5,6]. Hereditary thrombophilia has gained considerable attention as a risk factor for LCPD, although there are still controversies about this association or its potential implication in the severity of the disease [7,8]. Familial osteonecrosis of the femoral head has been associated with variant mutations of collagen type II [9,10]; however, the extracellular matrix might have a role in the pathogenesis or outcomes of LCPD. The thrombophilic factors include platelet transmembrane receptors that facilitate cell adhesion within extracellular matrix ligands; therefore, we aimed to investigate the possible correlation between the severity of LCPD and the presence of prevalent and well-known thrombophilic polymorphisms implied in extracellular matrix and tissue remodeling.

Our LCPD cohort and a group of controls were characterized for: (1) the PlA2 variant (rs5918(C)) of the platelet membrane glycoprotein (GP) IIb/IIIa, a central molecule in platelet adhesion and aggregation by the high-affinity binding of fibrinogen; (2) the rs1801133(T) variant in the *MTHFR* gene, playing a role involving tissue homocysteine in endothelial cell injury and adverse extracellular matrix remodeling; (3) the rs6025(A) variant encoding the Leiden mutation; (4) the rs1799963(A) variant of the prothrombin *F2* gene; the last two variants are associated with a prothrombotic genetic status. LCPD severity at the time of diagnosis is defined by Catterall classification based on the radiographic appearance of the epiphysis and metaphysis visible in osteonecrosis of the femoral head [11,12].

## 2. Materials and Methods

### 2.1. Patients

A cohort study was conducted, including 41 children diagnosed with LCPD disease in the Child Orthopedics Unit of the University Hospital Marqués de Valdecilla between January 1996 and December 2014. All patients presented clinical and radiological characteristics of LCPD. In 10 of them, the involvement was bilateral, representing a total of 51 hips.

Informed consent was obtained from all the individual participants (from their parents or guardians) according to the criteria of the Declaration of Helsinki, a statement of ethical principles for medical research involving human subjects, including research on identifiable human material and data. The study was supervised and approved by the Ethical Committee of the authors’ University and University Hospital (CEIC-C: Comité Ético de Investigación Clínica de Cantabria). After obtaining informed consent from the parents of all patients, a blood sample was drawn for DNA analysis. None of the patients had any clinical or familial evidence of significant vascular diseases, such as severe atherosclerosis, arterial thrombosis, coronary disease, hypertension, renal diseases, ischemic heart diseases, cerebrovascular diseases or coexisting disorders that could be associated with the polymorphisms analyzed. The presence of other causes of osteonecrosis of the hip was a reason for exclusion, such as by the use of steroids, hypothyroidism, epiphyseal dysplasia, metabolic bone disease, sickle cell disease, lupus, hip fracture, or septic arthritis. Demographic variables were collected as well as the evolution of the disease.

### 2.2. Controls

A control group of 118 healthy children and adolescents with no previous history of thromboembolism and representative of the general population of our region was assembled. Blood samples were obtained from them after informed consent for the AVENA study (study of cardiovascular risk related to nutritional and lifestyle habits, financed by a grant from the Spanish government, 2002–2005, and approved by the Ethical Committee of the University Hospital (CEIC-C: Comité Ético de Investigación Clínica de Cantabria)).

### 2.3. Genetic Analysis

DNA extraction. A sample of 2 mL of peripheral blood in EDTA was obtained from all participants; DNA extraction and purification were performed using the modified method of Higuchi [13] and the samples were frozen at −20 °C until further analysis. Subsequently, all genetic polymorphisms were studied by polymerase chain reaction (PCR) amplification and analysis of the restriction fragments (RFLP) after digestion with the corresponding restriction enzyme, by the protocols described below.

Characterization of PlA2 polymorphism of the platelet membrane glycoprotein (GP) IIb/IIIa (rs5918(C)), using a modified protocol described by Weiss et al. [14]. PCR amplification of DNA fragments was performed with the following primers: 5′ TTC TGA TTG CTG GAC TTC TCT T 3′ and 5′ TCT CTC CCC ATG GCA AAG AGT 3′. The amplified fragments were digested within Msp I (Invitrogen, USA) and separated in 3% agarose MS-12 (Laboratorios Conda, Madrid, Spain). PlA1 alleles exhibited two fragments of 221 and 46 bp; PlA2 (mutated) alleles exhibited three fragments of 177, 46, and 44 bp.

Characterization of C677T methylenetetrahydrofolate reductase (*MTHFR*) polymorphism (rs1801133(T)). DNA fragments were amplified by PCR following the protocol described by Frosst et al. [15], using the primers 5′-TGA AGG AGA AGG TGT CTG CGG GA-3′ and 5′-AGG ACG GTG CGG TGA GAG TG-3′. The PCR fragments were digested within the restriction enzyme Hinf I (GE Healthcare Life Sciences, Uppsala, Sweden). After electrophoresis in agarose 3%, a single 198 bp fragment corresponded to the C allele, and 175 and 23 bp fragments corresponded to the T allele (mutated allele).

Characterization of factor V Leiden mutation in the *F5* gene associated with factor V Leiden thrombophilia (rs6025(A)) was conducted using a modified protocol described by Bertina et al. [16]. PCR amplification of DNA fragments was performed with the primers 5′ TGC CCA GTG CTT AAC AAG ACC A 3′ and 5′ TGT TAT CAC ACT GGT GCT AA 3′. The amplified fragments were digested within Mml I (New England Biolabs, Ipswich, MA, USA) and separated in 3% agarose MS-12 (Laboratorios Conda, Madrid, Spain). The normal alleles exhibited three fragments of 163, 67, and 37 bp, and the mutated alleles (Leiden mutation) exhibited two fragments of 200 and 67 bp.

Characterization of the prothrombin thrombophilia associated with G20210A polymorphism in the *F2* gene (rs1799963(A)) was conducted by using a modified protocol described by Poort et al. [17]. The following primers were used for the PCR amplification of DNA fragments: 5′-TCT AGA AAC AGT TGC CTG GC-3′ and 5′-ATA GCA CTG GGA GCA TTG AA*G C-3′; subsequently, the amplified fragments were digested within Hind III (GE Healthcare Life Sciences, Uppsala, Sweden) and separated in 3% agarose MS-12 (Laboratorios Conda, Madrid, Spain). The normal alleles exhibited three fragments of 345 bp, and the mutated alleles (Leiden mutation) exhibited two fragments of 322 and 23 bp.

### 2.4. Risk Haplotypes

Different genetic risk–haplotypes categories were defined by the combination of pairs of genotypes. Haplotypes related to thrombophilia but also affecting extracellular matrix remodeling were indicated by the combination of GP IIb/III and *MTHFR* genotypes, whereas haplotypes related just to a prothrombotic status were defined by the combination of genotypes for factor V Leiden and prothrombin.

High-risk thrombophilia and potential adverse extracellular matrix remodeling were considered in haplotypes homozygous for the PlA2 variant of GP IIb/III (PlA2/PlA2) or homozygous for the T allele at *MTHFR* C677T (TT). Low-risk haplotypes were associated with the absence of the PlA2 variant (PlA1/PlA1) combined with non-mutated homozygous haplotypes or with heterozygosity for C677T (C/C and C/T); intermediate-risk phenotypes were identified by the presence of one allele (heterozygous) for PlA2 combined with heterozygous or non-mutated haplotypes for C667 (C/T and C/C).

The only genetic haplotypes associated with a substantial risk of thrombosis were defined as follows: haplotypes with a homozygous mutation for both factor V Leiden and prothrombin G20210 (high risk); haplotypes with a homozygous mutation for factor V Leiden plus heterozygous (G/A) for prothrombin G20210A (intermediate risk); haplotypes with non-mutated homozygous AA for G20210A plus heterozygous for factor V Leiden (low risk) [18].

### 2.5. Catterall Stages Classification of Severity

Markers of severity and prognosis of the LCPD are established by Catterall classification, which is based on the radiographic appearance of the epiphysis and metaphysis of the femoral head. Stage I: bone absorption changes visible in the anterior aspect of the epiphysis of the femoral head, changes are visible best in frog leg lateral view, no sclerosis is seen; Stage II: further bone resorption with slight femoral head collapse in the anterior aspect of femoral head, sclerosis; Stage III: almost entire femoral head involved in collapse with characteristic head-within-head appearance, sclerosis; Stage IV: complete collapse of the femoral head with the flattening and formation of dense sclerosis; additional metaphyseal changes may be visible, sclerosis, posterior remodeling [11,12,19].

### 2.6. Statistical Analysis

Quantitative variables were expressed as the mean and standard deviation, and categorical variables as absolute values and percentages (%). Each polymorphism was statistically described, estimating the prevalence of each allele and each possible genotype (allele and genotype frequencies) or were defined under risk haplotypes or a combination of genotypes. The frequencies of each allele and possible genotype or haplotype were calculated with their 95% confidence intervals (CIs). Pearson’s chi-squared or Fisher’s exact test was used for the comparison of categorical variables. Student’s *t*-test or Mann–Whitney U test was used for the comparison of quantitative variables. The frequency of the genotypes for each polymorphism was compared between the different groups using the Hardy–Weinberg equilibrium with the Pearson’s chi-squared test. The odds ratios (ORs) and their 95% confidence intervals (CIs) were calculated to estimate the relationship of polymorphisms with disease severity in patients vs. controls using non-conditional binary logistic regression models. All the tests were bilateral, and differences were considered significant when *p* < 0.05. All statistical analyses were performed using SPSS software version 15 for Windows (SPSS Inc, Chicago, IL, USA).

## 3. Results

A total of 36 of 41 patients (88%) with LCPD were boys, and 5 patients (12%) were girls. At diagnosis, the mean age was 5.42 years (SD 2.17), with a minimum age of 2.04 years and a maximum age of 10.85 years. The mean length of follow-up was 75.5 months (SD 45.3).

No patient had a personal history of previous thrombotic events or evidence of thrombotic disease in their first-degree relatives. During the course of the disease, both hips were affected in 10 patients (24%), 90% of which were male (9 males and 1 female). For some analyses, a total of 51 hips with LCPD were considered.

Considering the grades of severity based on Catterall classification, at the time of diagnosis, 3 hips (6%) were grade I on the Catterall scale, 7 hips were grade II (14%), 21 hips were grade III (41%), and 20 hips were grade IV (39%). This implies that, at diagnosis, 41 hips (80%) were in a severe phase of the disease (Catterall grades III and IV).

The genotype and allelic frequency distributions in both patients and controls were as summarized in Table 1:

-PlA2 polymorphism of the platelet membrane glycoprotein (GP) IIb/IIIa (rs5918(C)). Similar distributions in both LCPD cases and controls were observed for alleles and genotypes; thus, no significant risk for LCPD can be attributed to the presence of this polymorphism. The prevalence of mutated alleles was 17.1% (95% confidence intervals 10.5 to 26.6) in LCPD patients and 14.4% (95% confidence intervals, 10.5 to 19.5) in controls. The odds ratio of the A2 allele versus the A1 allele was 1.22 (95% confidence intervals 0.6 to 2.4; p = 0.561). The odds ratio for LCPD in those who were homozygous or heterozygous for the mutated allele relative to subjects with a normal genotype was 1.3 (95% confidence intervals 0.6 to 2.8).-Mutation C677T of metilenetetrahydrofolate reductase (MTHFR), (rs1801133(T)). No significant differences in allele frequencies were detected between cases of LCPD and controls. The prevalence of mutated alleles was 32.9% (95% confidence intervals 23.7 to 43.6) in the group of LCPD hips and 36% (confidence intervals 95% 30.2 to 42.3) in the controls, with the odds ratio of T versus C being 0.9 (95% confidence intervals 0.4 to 1.4).-Factor V Leiden mutation in the F5 gene causing factor V Leiden thrombophilia (rs6025(A)). Most cases (97%) and controls (99%) had a normal genotype for G1691A mutation in the F5 gene. No cases of mutated homozygous haplotypes were identified, either in cases or in controls. However, a positive trend was observed between LCPD and the presence of factor V Leiden mutation, with the odds ratio of A versus G being 2.9 (95% confidence intervals of 0.2 to 46.9).-G20210A mutation of the prothrombin F2 gene (rs1799963(A)). All patients with LCPD and most of the controls (97%) were G/G homozygous for the G20210A mutation of prothrombin F2 gene, and only four of the controls were heterozygous (G/A) for the polymorphism. No homozygous AA cases were detected. The prevalence of mutated A alleles was 1.7% (95% confidence intervals 0.6 to 4.3) in the controls, whereas no mutated A alleles were detected in the cases, with an odds ratio of 0.3 (95% confidence intervals 0.01 to 5.7) for allele A with respect to the G allele.

### Genetic Haplotypes

The frequencies of the different haplotypes (low-, intermediate- and high-risk haplotypes) and the number of hips affected by LCPD are shown in Table 2.

As described in the Methods section, haplotypes were defined by the combination of the different GP IIb/III and *MTHFR* genotypes. In total, 12% of the hips (*n* = 6) corresponded to patients with high-risk thrombophilia plus potential adverse extracellular matrix remodeling; 27% of the hips (*n* = 14) corresponded to patients with intermediate risk, and 61% (*n* = 31) corresponded to patients with low risk. When comparing LCPD patients and controls, we found no significant association between the defined genetic haplotypes and the risk of suffering from the disease (odds ratio 1.2, 95% confidence intervals 0.7 to 1.7). However, when severity at diagnosis was considered for each haplotype, we found that 67% of the defined high-risk haplotypes for thrombophilia and potential adverse extracellular matrix remodeling were classified as Catterall IV stage, whereas just 29% of the intermediate haplotypes were classified as Catterall IV (*p* = 0.05) (Table 3).

None of the patients with LCPD or the controls were classified as having genetic haplotypes with a substantial risk of thrombosis (homozygous for factor V Leiden or homozygous for prothrombin G20210A).

As an observational study with no intervention, surgery decisions were not influenced by the presence/absence of the studied polymorphisms, but by age, gender, and progression. Thus, our study design did not allow us to perform a multivariant analysis in order to define the role of the polymorphisms in the outcome according to Stulberg classification. However, after a six-year follow-up, 20% of the patients (10 out of 51 hips) had a non-favorable outcome with Stulberg grades IV–V, even after surgery. According with our results, the presence of the high-risk haplotypes (homozygous for *MTHFR*) should be considered in diagnosing the severity of LCPD, as up to 50% (two out of four) of the patients had Stulberg IV–V outcomes (three out of four in children under 6 years old).

## 4. Discussion

It is generally accepted that the disruption of the blood supply to secondary ossification centers in the femoral epiphyses is a key element for avascular necrosis, resulting in deformity of the femoral head, although there are few studies that address the implication of inherited thrombophilic status in the severity of LCPD. There is also evidence that remodeling of the extracellular matrix may be important in processes linked to mineralization and necrosis of the femoral head, and some thrombophilic factors involved in tissue damage after ischemic processes can also be implied in extracellular matrix remodeling mechanisms, thus related to the severity of LCPD.

In the present study, we analyzed the potential association of different genetic polymorphisms with a demonstrated risk for thrombosis and a potential disturbed matrix remodeling with the risk and severity of LCPD. As well as promoting coagulation, platelets are active in fibrinolysis, wound healing, angiogenesis, and bone formation and remodeling [20], with the platelet membrane glycoprotein (GP) IIb/IIIa being a central molecule in platelet adhesion, aggregation, and clot-forming process. The PlA1/A2 (rs5918) polymorphism in the GPIIb/IIIa genes is a demonstrated genetic factor of prothrombotic tendency, along with commonly recognized inherited thrombophilic factors such as the C677T polymorphism of *MTHFR*, classically associated with ischemic events, factor V Leiden, and the G20210A mutation in the prothrombin gene.

We observed that the presence of the *PlA2* platelet polymorphism and/or the TT homozygous status for the C677T polymorphism of *MTHFR* are both genetic conditions associated with the severity of LCPD at the time of diagnosis defined by Catterall stages III and IV. In our cohort, we did not observe a significant association between the presence of any other of the prothrombotic haplotypes and the risk of developing LCPD. Only for factor V Leiden mutation did we observe a significant trend related to the risk of Perthes disease (odds ratio 2.9; 95% confidence intervals, 0.2–47.8). These results are concordant with the findings of a recent meta-analysis [8], showing factor V Leiden mutation as the only prothrombotic polymorphism significantly related to Perthes disease.

The potential clinical impact of PlA2 polymorphism on the platelet glycoprotein IIIa gene has been investigated in several diseases in which thrombus formation tissue and remodeling mechanisms are key pathogenic factors [21,22], although never in LCPD. GPIIb/IIIa is the central molecule in platelet adhesion and aggregation by the high-affinity binding of fibrinogen, and the isoform PlA2 is associated with conformational changes in GPIIb/IIIa, which might exert an effect on the platelets’ nanomechanics as well as on the signaling pathways by triggering thrombotic events [23].

Metabolic changes associated with the polymorphic variants of *MTHFR* may increase the incidence of LCPD or potentially alter the recovery patterns of the necrotic head. There is a proven relationship between the mutated variant of *MTHFR* and mild–moderate hyperhomocysteinemia, and it has been suggested that endothelial cell aggression may be the initial mechanism by which hyperhomocysteinemia causes thrombosis. This situation interferes with tissue remodeling; therefore, differences in *MTHFR* activity also have a potential association with remodeling of the extracellular matrix, contributing to the pathogenesis of LCPD. In our study, we found that the four children with homozygous mutations of *MTHFR* had a severe form of the disease (Stage IV Catterall).

We did not observe any significant relationship between the other studied prothrombotic polymorphisms and the risk or severity of LCPD. Moreover, although some authors found that homozygous forms of the factor V Leiden mutation may play some role in the clinical course of LCPD, particularly in the more severe forms, we did not observe this association in our study [24].

The extent of epiphyseal involvement is one of the prognostic factors in LCPD. The greater the extent of the epiphyseal lesion, the greater the likelihood of collapse and deformity, and the worse the long-term outcomes (early osteoarthritis). The Catterall radiological classification of the extent of the disease continues to present it as the most important predictor of evolution for patients classified in groups III and IV [12]. However, although it has been used as a marker of severity and prognosis of the disease, it is questionable whether Catterall classification is clinically useful for making treatment decisions for LCPD patients. All patients in our LCPD cohort were classified and followed up by the same specialist. We found that the occurrence of at least one PlA2 allele or two alleles (homozygote) for the C677T polymorphism of *MTHFR* presented a significantly increased risk of having Catterall III or IV LCPD (*p* = 0.05); thus, their screening in at-risk children might be useful in the future as prognosis markers. In our cohort, all LCPD patients homozygous for the C677T mutation of *MTHFR* had a severe form at diagnosis (Catterall group IV).

As an observational study with no intervention, surgery decisions were not influenced by the presence/absence of the studied polymorphisms but by age, gender, and progression. However, we observed that according with our results, the presence of the high-risk haplotypes (homozygous for *MTHFR*) should be considered in the outcome of LCPD, especially for children under 6 years of age).

It is likely that in the development of LCPD, different etiopathogenic pathways simultaneously intervene; however, our findings, although limited by the sample size, rule out a significant role for the genetic thrombophilic status. However, severity at diagnosis is associated with the presence of polymorphisms disturbing extracellular matrix remodeling. Precisely determining the pathways that trigger the avascular necrosis process will be relevant for the early detection of LCPD, for the development of possible biomarkers, and for the implementation of medical therapeutic strategies that, together with the appropriate surgical attitude, will offer the best functional results for children affected by LCPD. However, our results are concordant with our initial hypothesis. Multicenter studies with larger sample sizes will be necessary to confirm our results.

## 5. Conclusions

Our results showed that inherited thrombophilic variants are not significant risk factors for LCPD but are associated with different levels of disease severity; this was observed, in particular, for those polymorphisms related with platelet adhesion and aggregation, cell injury, and adverse extracellular matrix remodeling. Subjects with an rs1801133(T) variant in *MTHFR* and an rs5918(C) variant encoding the A2 allele of the platelet glycoprotein displayed an increased risk of more severe forms of LCPD in the acute phase (Catterall III–IV). Being homozygous for the rs1801133(TT) variant or just having one copy of the rs5918(C) variant appeared to be associated with an increased risk for more severe forms of LCPD (Catterall groups IV and III) in the acute phase. The presence in homozygosis of the TT variant in *MTHFR* is suggested to be a risk factor for severe outcome (Stulberg IV–V after 6-year follow-up). Future multicenter studies would allow us to evaluate the association between LCPD and mechanisms of platelet activation and the release of substances into the extracellular space, which are determinants of tissue damage and repair mechanisms and are thus implicated in the pathogenesis of LCPD.

## Figures and Tables

**Table 1 children-08-00614-t001:** Genotype and allele distribution among LCPD hips and controls.

	Genotypes/Alleles	Controls	LCPD	OR (CI95%)	*p*
Factor V	GG	117 (99.2%)	49 (96.1%)	1	0.206
	GA	1 (0.8%)	2 (3.9%)	4.8 (0.4 to 53.9)	
	AA	0	0		
Allele frequencies	G	235 (99%)	100 (98%)		0.452
	A	1 (1%)	2 (2%)	4.7 (0.4 to 52.4)	
*MTHFR*	CC	47 (39.8%)	26 (51%)	1	0.181
	CT	57 (48.3%)	20 (39.2%)	0.6 (0.3 to 1.2)	
	TT	14 (11.9%)	5 (9.8%)		
Allele frequencies	C	151 (63.9%)	72 (70.6%)		0.290
	T	85 (36.1%)	30 (29.4%)	0.7 (0.4 to 1.2)	
GP IIb/IIIa	A1A1	87 (73.7%)	34 (66.7%)	1	
	A1A2	28 (23.7%)	16 (31.4%)	1.4 (0.7 to 2.9)	0.351
	A2A2	3 (2.5%)	1 (1.9%)		
Allele frequencies	A1	202 (85.6%)	84 (82.4%)		
	A2	34 (14.4%)	18 (17.6%)	1.4 (0.7 to 2.4)	0.552
Prothrombin	GG	114 (96.6%)	51 (100%)	1	
	GA	4 (3.4%)	0 (0%)	0.000	0.999
	AA	0 (0%)	0 (0%)		
Allele frequencies	G	232 (98.3%)	102 (100%)		
	A	4 (1.7%)	0	0.3 (0.01 to 4.7)	0.438

Abbreviations: GP IIb/IIIa: platelet glycoprotein IIb/IIIa; LCPD: Legg–Calvé–Perthes disease; MTHFR: methylenetetrahydrofolate reductase; OR: odds ratio; *p*: *p*-value.

**Table 2 children-08-00614-t002:** Haplotype distributions among the LCPD hips analyzed.

Number of Evaluated LCPD Hips = 51					
Low Risk * *n* = 31		Intermediate Risk * *n* = 14		High Risk * *n* = 6	
A1A1–CC	15 (48.4%)	A1A2–CT	3 (21.4%)	A2A2–TT	0
A1A1–CT	16 (51.6%)	A1A2–CC	11 (78.6%)	A1A2–TT	2 (33.3%)
				A1A1–TT	3 (50%)
				A2A2–CT	1 (16.7%)
				A2A2–CC	0

* Low-risk haplotypes: a combination of normal homozygous status for the PlA2 (A1A1) polymorphism with homozygous normal (CC) and heterozygous (CT) status for the C677T mutation of the *MTHFR* gene. Intermediate-risk haplotypes: those heterozygous for the PlA2 (A1A2) polymorphism plus normal homozygous (CC) or heterozygous (CT) haplotypes for the C677T mutation of the *MTHFR* gene. High-risk haplotypes: a combination of the homozygous status for *MTHFR* (TT) or *PlA2* (A2A2) genes.

**Table 3 children-08-00614-t003:** Haplotype distributions and the severity of LCPD hips based on Catterall classification.

	Haplotypes			
	Low Risk *	Intermediate Risk *	High Risk *	*p*
Catterall I	1 (3.2%)	1 (7.1%)	1 (16.7%)	0.05
Catterall II	2 (6.5%)	5 (35.7%)	0	
Catterall III	16 (51.6%)	4 (28.6%)	1 (16.7%)	
Catterall IV	12 (38.7%)	4 (28.6%)	4 (66.6%)	

* Low-risk haplotypes: a combination of normal homozygous status for the PlA2 (A1A1) polymorphism with normal homozygous (CC) and heterozygous (CT) status for the C677T mutation of *MTHFR*. Intermediate-risk haplotypes: heterozygous for the PlA2 (A1A2) polymorphism plus normal homozygous (CC) or heterozygous (CT) for the C677T mutation of the *MTHFR* gene. High-risk haplotypes: a combination of homozygous *MTHFR* (TT) or *PlA2* (A2A2).

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
