# Peer review of "PlA2 Polymorphism of Platelet Glycoprotein IIb/IIIa and C677T Polymorphism of Methylenetetrahydrofolate Reductase (MTHFR), but Not Factor V Leiden and Prothrombin G20210A Polymorphisms, Are Associated with More Severe Forms of Legg–Calvé–Perthes Disease"

_children, 2021, doi:10.3390/children8070614_

Round 1

Reviewer 1 Report

The paper has so many abbreviations that it becomes an alphabet soup of abbreviations. Some like odds ratio OR are also abbreviated O.R. in the abstract. I would use the words odds ratio and confidence intervals in the text instead of abbreviating them except in the table. For the table please expand the long-form of all abbreviations below the table.

The study follows a group of children diagnoses from 1996 to 2014 a 19 year period of time. The authors chose to use the Catterall classification as the measure of severity of Perthes. While this is certainly a good way to assess the severity of disease during the acute phase it does not necessarily correlate with the final shape of the femoral head which is the most important prognostic criterion. Since the authors have such long term followup of this group of patients why did they not report or analyze the final shape of the femoral head according to the Stulberg classification and try and correlate that with the thrombophilic factors they studied. If this can be added to this paper it will make it the best study on the association of thrombophilia and true severity of the disease. I recommend this paper be published with or without the Stulberg data, but highly encourage the authors to try and include as much Stulberg data as possible to this paper.

Author Response

Response to Reviewer 1 Comments

Point 1: The paper has so many abbreviations that it becomes an alphabet soup of abbreviations. Some like odds ratio OR are also abbreviated O.R. in the abstract. I would use the words odds ratio and confidence intervals in the text instead of abbreviating them except in the table. For the table please expand the long-form of all abbreviations below the table.

Response 1: Thank you very much for your kind suggestion and apologizes for the soup of abbreviations. We already changed the O.R. in the abstract and also suppress some abbreviations in the text (i.e. ECM, OD, CI).

We included the long-form of all abbreviations below the table 1.

Point 2: The study follows a group of children diagnoses from 1996 to 2014 a 19 year period of time. The authors chose to use the Catterall classification as the measure of severity of Perthes. While this is certainly a good way to assess the severity of disease during the acute phase it does not necessarily correlate with the final shape of the femoral head which is the most important prognostic criterion. Since the authors have such long term followup of this group of patients why did they not report or analyze the final shape of the femoral head according to the Stulberg classification and try and correlate that with the thrombophilic factors they studied. If this can be added to this paper it will make it the best study on the association of thrombophilia and true severity of the disease. I recommend this paper be published with or without the Stulberg data, but highly encourage the authors to try and include as much Stulberg data as possible to this paper.

Response 2: We completely agree with your comment, Catterall classification just assess the severity during the acute phase and do not exactly correlate with the final shape of femoral head according Stulberg classification. However we are not able to extract conclusions in this sense. Most of patients with Stulberg IV-V underwent surgery treatments and were Catteral III-IV in the acute phase. Surgical decisions were based according age, gender, outcome, thus it is not possible to correlate with the genotype for the studied polymorphisms.  Although we included a significant number of hips (51) we were not able to perform an accurate  multivariant analysis in order to define the role of the polymorphism over the age. Moreover, this is not the aim of our study but we did not find a significant statistical correlation between Catteral and Stulberg when the age was not consider. However, all patients with Stulberg IV-V were Catteral III-IV and 8 out 9 underwent surgery.

The haplotypes of risk influenced the outcome same as age. The only patient with Stulberg V and Catteral III was 15 years old at diagnosis, whereas the childeen under 6 years old with Stulberg V (Catterall IV) has an haplotype of risk.  According gender,  girls with the high risk haplotipe have worst outcome than those with low risk

We included the following text in the manuscript: Our study design do not allow us to perform a multivariant analysis in order to define the role of the polymorphisms in the outcome according to Stulberg classification. As an observational study with no intervention, surgery decisions were not influenced by the presence/absence of the studied polymorphisms but for age, gender and progression. After a six year follow-up, 20% of patients (10 out of 51 hips) have a non-fauvorable outcome with Stulberg grades IV-V, even undergoing surgery. According with our results, the presence of the high-risk haplotypes (homocygous for MTHFR) should be considered in the outcome of LCPD, as up to 50% (2 out of 4) have Stulberg IV-V (3 out of 4 in children under 6 years old).

Reviewer 2 Report

i think is  good research and very interesting. i would like to improve the conclusions that appear a little poor in content

Author Response

We include the english revision certificate.

Point 1.- I think is  good research and very interesting.

Response 1: Thank you very much for your comments

Point 2.- I would like to improve the conclusions that appear a little poor in content

Response 2.- Thank you very much for your input. According your suggestion, we improved the quality of content in the conclusions (see text). 
